# Baseline Serum Biomarkers Predict Response to a Weight Loss Intervention in Older Adults with Obesity: A Pilot Study

**DOI:** 10.3390/metabo13070853

**Published:** 2023-07-17

**Authors:** David H. Lynch, Blake R. Rushing, Wimal Pathmasiri, Susan McRitchie, Dakota J. Batchek, Curtis L. Petersen, Danae C. Gross, Susan C. J. Sumner, John A. Batsis

**Affiliations:** 1Division of Geriatric Medicine and Center for Aging and Health, University of North Carolina at Chapel Hill, BMBS 5003 Old Clinic/CB #7550, Chapel Hill, NC 27599, USAjohn.batsis@unc.edu (J.A.B.); 2Nutrition Research Institute, Department of Nutrition, University of North Carolina, Kannapolis, NC 28081, USAsusan_sumner@unc.edu (S.C.J.S.); 3Department of Nutrition, Gillings School of Global Public Health, University of North Carolina at Chapel Hill, Chapel Hill, NC 27599, USA; 4Geisel School of Medicine, The Dartmouth Institute for Health Policy, Hanover, NH 03755, USA

**Keywords:** older adults, weight loss intervention, biomarkers

## Abstract

Caloric restriction and aerobic and resistance exercise are safe and effective lifestyle interventions for achieving weight loss in the obese older population (>65 years) and may improve physical function and quality of life. However, individual responses are heterogeneous. Our goal was to explore the use of untargeted metabolomics to identify metabolic phenotypes associated with achieving weight loss after a multi-component weight loss intervention. Forty-two older adults with obesity (body mass index, BMI, ≥30 kg/m^2^) participated in a six-month telehealth-based weight loss intervention. Each received weekly dietitian visits and twice-weekly physical therapist-led group strength training classes with a prescription for aerobic exercise. We categorized responders’ weight loss using a 5% loss of initial body weight as a cutoff. Baseline serum samples were analyzed to determine the variable importance to the projection (VIP) of signals that differentiated the responder status of metabolic profiles. Pathway enrichment analysis was conducted in Metaboanalyst. Baseline data did not differ significantly. Weight loss was 7.2 ± 2.5 kg for the 22 responders, and 2.0 ± 2.0 kg for the 20 non-responders. Mummichog pathway enrichment analysis revealed that perturbations were most significant for caffeine and caffeine-related metabolism (*p* = 0.00028). Caffeine and related metabolites, which were all increased in responders, included 1,3,7-trimethylxanthine (VIP = 2.0, *p* = 0.033, fold change (FC) = 1.9), theophylline (VIP = 2.0, *p* = 0.024, FC = 1.8), paraxanthine (VIP = 2.0, *p* = 0.028, FC = 1.8), 1-methylxanthine (VIP = 1.9, *p* = 0.023, FC = 2.2), 5-acetylamino-6-amino-3-methyluracil (VIP = 2.2, *p* = 0.025, FC = 2.2), 1,3-dimethyl uric acid (VIP = 2.1, *p* = 0.023, FC = 2.3), and 1,7-dimethyl uric acid (VIP = 2.0, *p* = 0.035, FC = 2.2). Increased levels of phytochemicals and microbiome-related metabolites were also found in responders compared to non-responders. In this pilot weight loss intervention, older adults with obesity and evidence of significant enrichment for caffeine metabolism were more likely to achieve ≥5% weight loss. Further studies are needed to examine these associations in prospective cohorts and larger randomized trials.

## 1. Introduction

The rise in global obesity, coupled with the rise in the aging population, has resulted in a larger proportion of older adults with obesity [1]. Specifically, an estimated 35% of older adults (>65 years) in the U.S. are living with obesity [2]. Furthermore, age-related body composition changes, and decline in physical activity, lead to a decrease of energy expenditure, all of which are associated with redistribution of fat mass—particularly in the abdomen—and obesity [1]. Abdominal obesity in older adults is associated with an increased risk of multimorbidity, nursing home placement, disability, and death [2,3,4]. Older adults with obesity are at a higher risk of frailty, which can progress to sarcopenic obesity, which, in turn, increases mortality, exacerbates functional decline, and decreases quality of life [5]. Dietary and multi-modal exercise interventions are safe and effective in promoting weight loss and improving physical function in older adults with obesity [2,6]. However, because of unique physiological, biological, and behavioral traits, there is heterogeneity in individual responses to various interventions [7,8]. Given the multiple factors associated with obesity and the variability in responses to interventions, the most effective treatments for older adults with obesity are those that recognize and adapt to each individual. Specifically, individually crafted programs designed uniquely for each patient seem to be the most effective for this population [8].

Precision medicine can be effective at identifying and addressing the heterogeneity in intervention responses among individuals with obesity [9]. It is built upon the principle of delivering the right treatment to the right person at the right time [10]. Due to the prevalence of multimorbidity in the elderly population, precision medicine has the potential to be of great benefit [11]. Furthermore, obesity has a wide range of factors associated with its onset, such as genetics, lifestyle, and metabolic profile. Thus, utilizing precision medicine’s “deep phenotyping” approach shows promise in promoting weight loss in older adults with obesity, as it allows for personalized treatments to one’s specific metabolic makeup and multimorbidity—as opposed to a one-treatment-fits-all approach [12]. Multi-omics analyses combine multiple forms of biological data (e.g., metabolomics, proteomics) to analyze complex biological pathways and, thus, is an essential tool within the field of precision medicine [13]. Combining the principles of precision medicine with the multi-omics approach may unveil the subtle phenotypical characteristics responsible for an individual’s response to an intervention [12,13]. Metabolomics is the emerging field of precision medicine that involves analyzing an individual’s composition of thousands of metabolites and relating the data to metabolic and physical changes/responses [14]. For instance, many researchers use metabolomics to study the metabolic response to exercise in both the sport and clinical fields [15]. Metabolomics has also played a key role in disease progression studies. One study incorporated metabol- and lipid-omics in assessing the interaction between lipids and metabolites in pathogenesis [16].

In the current study, we aimed to use metabolomic strategies and precision medicine principles to disentangle the heterogeneity in responses to a weight loss intervention in older adults with obesity. Specifically, we sought to use untargeted metabolomics to identify metabolic phenotypes that were associated with response to the intervention. We conducted a non-randomized, single-arm study at a community ageing center on adults ≥ 65 years or older with a BMI of 30 kg/m^2^ or higher [17]. Participants were visited weekly by a dietician and participated in twice-weekly strength/flexibility/balance sessions led by a physical therapist. An aerobic activity regimen was prescribed to be conducted at home and monitored using self-report logs. Each participant received a Fitbit Alta HR, which provided objective step counts. We anticipate that this approach may lay the foundation for targeted lifestyle interventions for older adults with obesity.

## 2. Methods

### 2.1. Study Design 

We have previously outlined details on study participants, recruitment, setting, and design [17]. In brief, this was a pilot, pre/post, 26-week technology-based weight loss intervention, which included exercise and nutrition components. We recruited 53 older adults (>65 years) who were community-dwelling at baseline. The intervention consisted of 25 virtual nutrition sessions with a dietician and fifty exercise sessions led by a physical therapist. Based on established evidence of impact on long-term clinical outcomes, those with 5% loss in baseline body weight or more were classified as responders, and those with <5% were classified as non-responders. 

### 2.2. Metabolomics Sample Preparation

Serum samples (50 µL) were mixed with 400 µL of 80% methanol (Fisher Scientific, Waltham, MA, USA), containing 500 ng/mL tryptophan-d5, as an internal standard, and was vortexed by a multiple-tube vortex mixer for 2 min at 5000 rpm at room temperature. The same volume (50 µL) of LC-MS grade water was used for blank samples, and the same volume (50 µL) of National Institute of Standards and Technology (NIST) reference serum (SRM 909c) was used for external reference materials. Quality control study pool (QCSP) samples were made by pooling an additional 10 µL of each study sample into a singular mixture, which was distributed into 50 µL aliquots. Blanks, NIST reference aliquots, and QCSP aliquots were processed in an identical manner as the study samples. All samples were centrifuged at 16,000 rcf for 10 min at 4 °C. The resulting supernatant (350 µL) was then dried by a SpeedVac (Labconco, Kansas, MO, USA) overnight. Dried sample extracts were reconstituted with 100 µL of a water-methanol solution (95:5, *v*/*v*), vortexed for 10 min at 5000 rpm, and then centrifuged at 16,000 rcf for 10 min at 4 °C. The supernatant was transferred to pre-labeled autosampler vials and 5 µL was injected onto the LC-MS column for untargeted analysis. 

### 2.3. UHPLC-HRMS Data Acquisition

Metabolomics data were acquired on a Vanquish UHPLC system coupled to a Q Exactive™ HF-X Hybrid Quadrupole-Orbitrap Mass Spectrometer (Thermo Fisher Scientific, San Jose, CA, USA). Separation of metabolites was carried out using an HSS T3 C18 column (2.1 × 100 mm, 1.7 µm, Waters Corporation, Milford, MA, USA) at 50 °C with a binary mobile phase of water (A) and methanol (B), each containing 0.1% formic acid (*v*/*v*). The UHPLC linear gradient started at 2% B, and increased to 100% B in 16 min, then held for 4 min, with the flow rate at 400 µL/min. UHPLC-HRMS data was acquired in a mass range from 70 to 1050 m/z in positive mode, and the MS/MS fragmentation data was acquired under data-dependent acquisition mode using the 20 most abundant ions per scan. Quality control materials (QCSPs, NIST reference aliquots, and blanks) were interspersed amongst the study samples throughout the run sequence. 

### 2.4. Metabolomics Data Preprocessing

The UHPLC-HRMS data was processed by Progenesis QI (version 2.1, Waters Corporation, Milford, MA, USA) for peak picking and alignment. Background signals were excluded by removing peaks with a higher mean intensity of the blanks, as compared to the QCSPs based on the unnormalized data. The data was normalized to a reference study pool sample using the “normalize to all” function in Progenesis, and signals with a relative standard deviation (RSD) > 50% across study pool replicates were excluded from further analysis [18]. 

### 2.5. Compound Identification and Annotation

Peaks detected by UHPLC-HR-MS were identified or annotated by matching to an in-house reference standard RT, Mass, and MS/MS library of over 2400 compounds run on the untargeted platform, or to public databases (NIST, METLIN, HMDB). To report the evidence basis for each metabolite match, an ontology system is given, based on matches by accurate mass (MS, <5 ppm), retention time (RT, ±0.5 min), and fragmentation similarity (MS/MS, >30). OL1 refers to an in-house library match by MS, MS/MS, and RT; OL2a refers to an in-house library match by MS and RT; OL2b refers to an in-house library match by MS and MS/MS; PDa refers to a public database match by MS and experimental MS/MS (NIST or METLIN); PDb refers to a public database match by MS and theoretical MS/MS (HMDB); PDc refers to a public database match by MS and isotopic similarity; PDd refers to a public database match by MS only. However, it is important to note that structural isomers with D and L configurations may not always have separate references on the untargeted platform used for comparison. 

### 2.6. Statistical Analysis

Multivariate analyses were conducted using SIMCA^®^ 16 (Sartorius Stedim Data Analytics AB, Umeå, Sweden). Normalized untargeted LCMS data from the baseline visit were mean-centered and UV-scaled prior to principal component analysis (PCA) and orthogonal partial least squares discriminant analysis (OPLS-DA). PCA is an unsupervised analysis that reduces dimensionality by projecting the data onto a new coordinate system that allows for visualizing the distribution of the data, including clustering [19]. The PCA scores plots of the data were inspected to ensure that the quality control study pools created from the individual study samples clustered near the center of all samples, and that the NIST reference material clustered, a quality control method that is widely used in metabolomic studies [20]. OPLS-DA is a supervised analysis for categorical outcomes; it was used to determine the peaks that were important for differentiating the responders to the intervention (≥5% weight loss) from the non-responders. Loadings plots and variable influence on projection (VIP) plots were inspected, and peaks that had a VIP ≥ 1.0 with a jack-knife confidence interval that did not include 0 were determined to be important for differentiating responders from non-responders. The VIP statistic summarizes the importance of the peak in differentiating the phenotypic group, with higher values indicating that a peak is more important in differentiating the phenotype [19]. The OPLS-DA model used a 7-fold cross-validation to assess the model’s predictive performance (Q^2^).

Statistical analyses were conducted using SAS 9.4 (SAS Institute Inc., Cary, NC, USA). Hypothesis tests for continuous variables were conducted using a two-sided *t*-test with the Satterthwaite correction for unequal variances. The chi-square test was used to test for differences in categorical variables, with the Fisher’s Exact test being used when a categorical variable had small cell counts (current smoker, education, income). In this exploratory metabolomics study, *p*-values < 0.05 were considered to be statistically significant and were not adjusted for multiple testing, since the study was not powered to a specific hypothesis [21,22].

### 2.7. Pathway Analysis

Pathway enrichment was conducted using the Mummichog algorithm in the “Functional Analysis” module in Metaboanalyst 5.0 [23,24]. All features (*m/z*) remaining after filtering the data were entered together with the mass-to-charge ratio (*m/z*), retention time, *p*-value, and fold change between the comparison of responders and non-responders at baseline. A *p*-value cut-off of 0.05 was used for the size of the permutation group that the algorithm used for selecting significant features for metabolite matching. A 3-ppm tolerance was used for mass accuracy for annotating peaks to metabolites and identifying candidate pathways. All possible metabolites that were matched by m/z were searched in the Homo sapiens (human) [MFN] pathway library. Significance is reported as both uncorrected (FET) and corrected (Gamma) *p*-values.

## 3. Results

After preprocessing, 9647 peaks remained in the untargeted metabolomics dataset. Supervised OPLS-DA analysis of baseline metabolomics data for responders to weight loss in the clinical trial versus those who were non-responders showed significant separation between the two groups with a high goodness-of-fit value for the model (R2Y = 0.905), although it had a low predictive ability (Q2 = −0.16) (Figure 1), presumably due to the low sample size. Notably, the unsupervised PCA analysis (Appendix A) did not reveal a good visual separation between responders and non-responders. However, the supervised analysis was used to determine the signals/metabolites that contribute to the separation of responders and non-responders in the OPLS-DA model, allowing for observable metabolic differences between the two groups. The variable importance to projection (VIP) score provides an estimate of the importance of each of the signals/metabolites to the differentiation of the responders and non-responders in the OPLS-DA model. A VIP ≥ 1 indicates that the signal/metabolite is important to the differentiation of the responders and non-responders (Appendix A). No significant differences were observed in any clinical baseline characteristics between responders and non-responders (Table 1), indicating that metabolic differences were attributable to responder status in this analysis.

Statistical analysis of the preprocessed data set revealed 697 peaks with *p* < 0.1 between responders and non-responders (Appendix A). To determine metabolic pathways differentiating the two groups (responder versus non-responder), pathway enrichment analysis using the Mummichog algorithm was performed using all 9647 peaks in the untargeted dataset (Figure 2, Table 2). Seven metabolic pathways were significantly different between the two groups (*p* < 0.05). Significant metabolic pathways included (1) caffeine metabolism; (2) valine, leucine, and isoleucine degradation; (3) lysine metabolism; (4) galactose metabolism; (5) starch and sucrose metabolism; (6) hexose phosphorylation; and (7) pentose phosphate pathway. Caffeine metabolism was notably the most significantly perturbed pathway (*p* = 0.00049, Table 2). The full list of pathway analysis results is presented in Appendix A.

To gain a better understanding of metabolic differences between responders and non-responders, untargeted metabolomics peaks were matched to an in-house library of standards and public databases. A number of 645 and 4748 peaks were matched to the in-house library and to public databases, respectively. Of the in-house matches, 57 serum metabolites had a *p* < 0.1 between responders and non-responders (Appendix A). Included in these were caffeine (1,3,7-trimethylxanthine (VIP = 2.0, *p* = 0.033, FC = 1.9) and its metabolites, theophylline (VIP = 2.0, *p* = 0.024, FC = 1.8), paraxanthine (VIP = 2.0, *p* = 0.028, FC = 1.8), 1-methylxanthine (VIP = 1.9, *p* = 0.023, FC = 2.2), 5-acetylamino-6-amino-3-methyluracil (VIP = 2.2, *p* = 0.025, FC = 2.2), 1,3-dimethyl uric acid (VIP = 2.1, *p* = 0.023, FC = 2.3), and 1,7-dimethyl uric acid (VIP = 2.0, *p* = 0.035, FC = 2.2). All caffeine metabolites were matched at a level of OL1 (RT, MS, and MS/MS match), and were all increased in responders (Table 3). The distribution of caffeine metabolites showed that differences between responders and non-responders was not due to a small number of outlier measurements (Figure 3). The variation in the level of caffeine metabolites across individuals within a group may be related to inter-individual differences in the genetics of caffeine metabolism, the rate of caffeine metabolism, and/or the time since last caffeine consumption.

Analysis of additional metabolites, matched to the in-house library at a level of OL1 or OL2a, showed differences in endogenous metabolic pathways (Table 4). Responders versus non-responders had perturbations in amino acid/peptides, lipid/fatty acids, carbohydrates, nucleic acids, and a form of vitamin D. Additionally, differences were found in phytochemical and microbiome-related metabolites, the majority of which were increased in the baseline serum of individuals who has a significant weight loss. Acetaminophen was significantly reduced in responders, indicating a lesser need for pain management. Monoethyl phthalate was increased in responders, potentially derived from consuming coffee [25].

## 4. Discussion

In this pilot study of a weight loss intervention in older adults with obesity, we found that responders in the weight loss clinical trial had differences in baseline metabolomic profiles compared to non-responders. Specifically, caffeine (1,3,7-trimethylxanthine) and caffeine-related metabolites (paraxanthine, theobromine, 1-methylxanthine,1,3-dimethyl uric acid, and 1,7-dimethyl uric acid) were all increased in the baseline samples of responders compared with non-responders (Figure 4). This data elucidates a correlation between older adults with obesity losing ≥5% of their body weight and increased caffeine intake or altered metabolism.

Our findings, with respect to caffeine metabolism and weight loss, are consistent with existing literature. A systematic review of the effects of caffeine intake on weight loss showed a correlation between caffeine intake and BMI/fat mass reduction [26]. A randomized control trial in younger men showed caffeine ingestion to be associated with increased energy expenditure and increased lipid turnover [27]. Furthermore, in a randomized control trial of a diet and exercise intervention conducted in competitive cyclists, caffeine intake was associated with increased lipid metabolism and a decrease in perceived exertion, enabling more exercise to be performed [28]. This is consistent with an increase in lipids and fatty acids in responders (compared with non-responders) in our study. Therefore, in our study, it is possible that caffeine metabolism could have altered the effectiveness of exercise in responders, thus enabling them to lose more weight, while also enhancing biological pathways that further promoted weight loss [27,28,29].

While our findings in this small pilot study suggest a potential relationship between caffeine metabolism and weight loss in obese older adults, future studies are needed to further disentangle the relationship between caffeine consumption, caffeine metabolism, exercise capacity, and weight loss. Such studies could provide information as to whether caffeine itself, or genetic factors related to the rate of caffeine metabolism, are associated with weight loss in this population [30,31]. Future studies can use these results to inform targeted genetic and caffeine metabolism analyses to support our findings. This will allow for further characterization of metabolites that are associated with both ingestion and exercise response. This study has the potential to inform precision medicine-based weight loss interventions. However, more information is needed to determine whether caffeine may be beneficial in all individuals, or only those with specific polymorphism in caffeine metabolism. Understanding the effects of caffeine and its relationship with caffeine metabolism can help physicians make decisions about whether to prescribe caffeine as a weight loss treatment. Furthermore, based on one’s genetic polymorphisms and caffeine metabolism, physicians can perhaps make better decisions regarding the intensity of the exercise prescribed.

Additional differences were found in energy-producing metabolic pathways between responders and non-responders. This indicates that macronutrient metabolism may differ between responders and non-responders, and that basal protein, lipid, and/or carbohydrate metabolism may predispose an individual to be more responsive to weight loss interventions. Additionally, caffeine has previously been reported to alter substrate utilization for energy production, giving rise to the possibility that these differences may be a direct consequence of the differences in caffeine metabolism seen in responders [32]. Furthermore, our finding that increased phytochemical and microbiome-related metabolites were mostly increased in responders suggests that a phytochemical-rich diet and microbiome status may play a role in determining responder status. Several of these metabolites belong to the metabolism of pipecolic acid, which is metabolically produced by the intestinal microbiota from lysine [33]. Further, 3-hydroxyhippuric acid and dihydroferulic acid are indicators of metabolism of phenolic compounds and have been shown to arise following coffee consumption; both increased in responders. N-acetyl-beta-alanine, an amino acid derivative, has also been found in coffee sources and was also increased in responders in the current study [34,35]. In addition, dimethylglycine (a metabolite of the vitamin-like compound choline) was significantly increased in responders. More research is needed to determine if elevated levels of these compounds play a functional role in modulating response to weight loss interventions.

Strengths of this study include its use of novel omics and precision medicine approaches that have, thus far, been underutilized in older adults with obesity. These approaches may be particularly useful in this population, given the heterogeneity present in phenotypes and response to interventions [7,9]. Identifying how a single population differs on the premises of biological markers can be used in precision medicine to provide specific treatment that may benefit a certain group at an enhanced level compared to others; altogether, this could help make weight loss interventions for older adults more individualized [9]. There are several significant limitations of this study. Most notably, our small sample size in this pilot study greatly limits the generalizability and conclusions that can be drawn from these results. Additionally, the untargeted methods used in this study were exploratory in nature and further validation with targeted methods should be performed to better understand the role of these metabolites in weight loss interventions. Additionally, caffeine and dietary intake of the participants were unknown during the time of intervention. Therefore, it is hard to draw conclusions about the effects of caffeine versus caffeine metabolism on weight loss in older adults with obesity. Lastly, specific information about capacity of participants for physical activity is unknown, limiting our ability to understand if weight loss in responders was related to their capacity to be able to exercise more than non-responders.

In conclusion, our data suggests that older adults with obesity losing ≥5% of their body weight in response to a lifestyle intervention may have distinct metabolic phenotypes at baseline and may have a different caffeine metabolism than those who fail to respond. However, more studies need to be conducted in larger prospective cohorts. Additionally, future studies on the feasibility of metabolomic analysis to predict exercise response are also needed to identify common trends among multiple studies, enabling better conclusions to be drawn.

## Figures and Tables

**Figure 1 metabolites-13-00853-f001:**
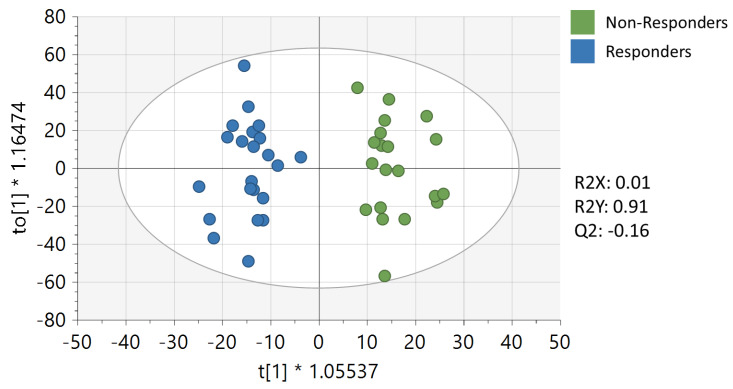
Orthogonal partial least squares discriminant analysis on baseline samples.

**Figure 2 metabolites-13-00853-f002:**
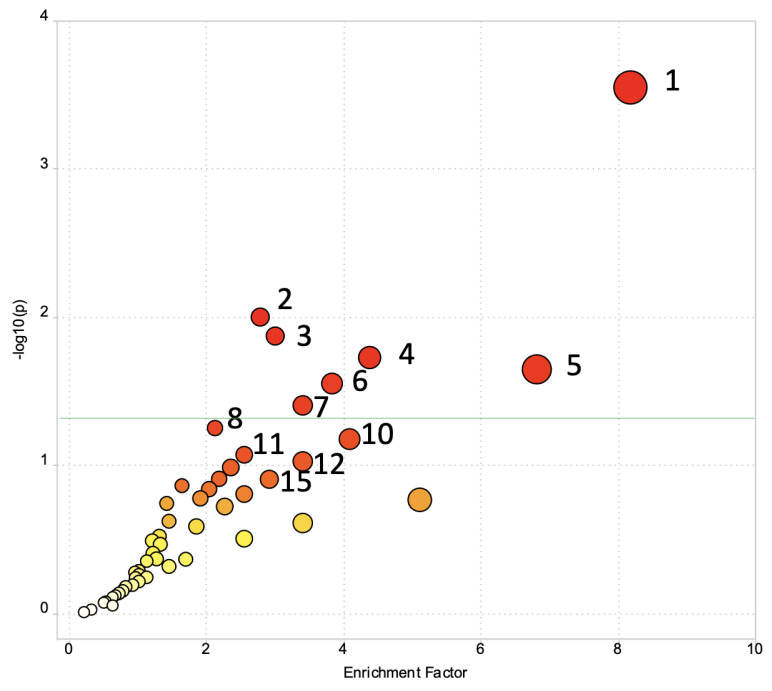
Pathway enrichment analysis of baseline serum samples comparing responders versus non-responders. Statistical significance is at approximately 1.3 on the −log10 scale (denoted by a green line). Pathway names corresponding to annotated numbers can be found in Table 2.

**Figure 3 metabolites-13-00853-f003:**
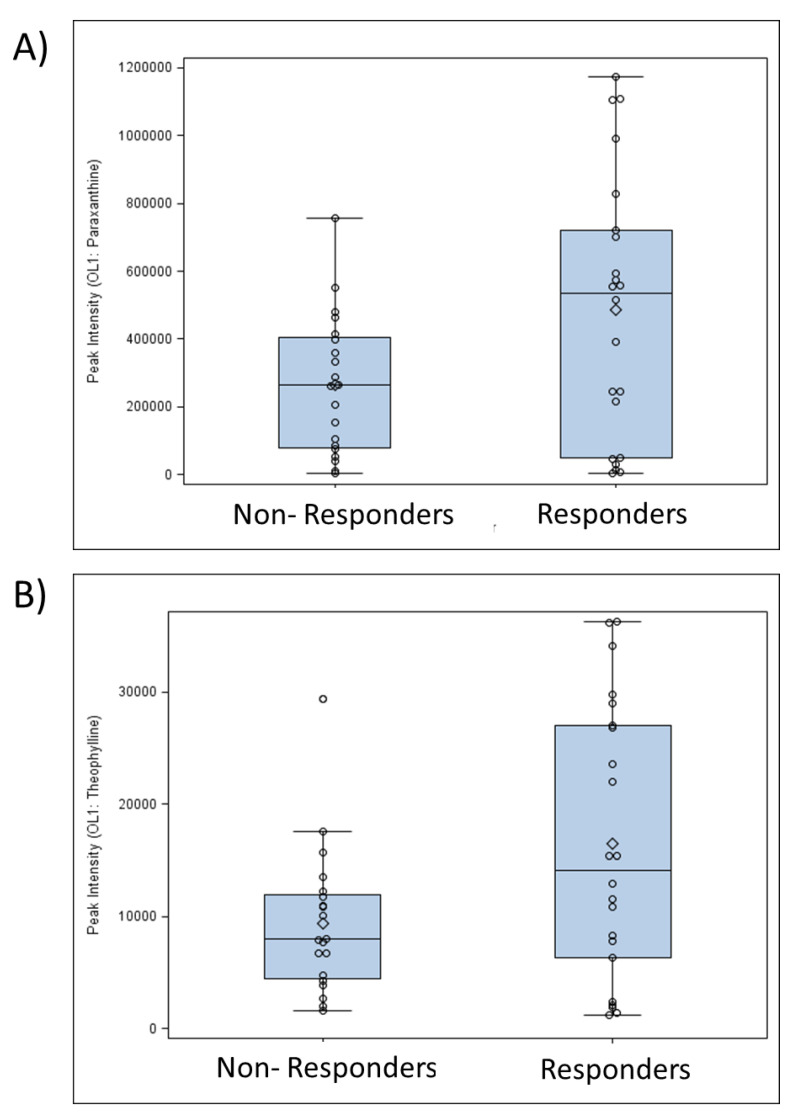
Distribution of (**A**) paraxanthine and (**B**) theophylline in responders and non-responders. Individual study participants are indicated by circles, the mean is indicated by the diamond within each box, and the median is represented by a horizontal line within each box. The bottom of the box represents the 25th percentile and the top of the box represents the 75th percentile.

**Figure 4 metabolites-13-00853-f004:**
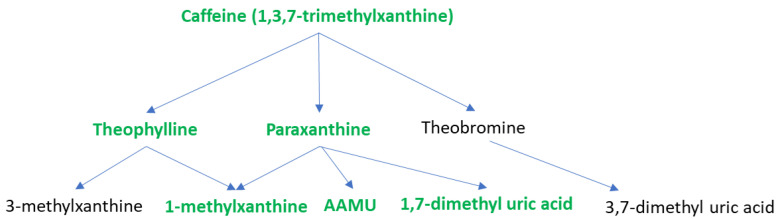
Caffeine and caffeine metabolites differed in baseline samples between responders and non-responders. AAMU, 5-Acetylamino-6-amino-3-methyluracil. The metabolomics method did not differentiate between the structural isomers depicted. Metabolites in green were significantly increased in responders.

**Table 1 metabolites-13-00853-t001:** Subject Characteristics.

Subject Characteristic at Baseline	Weight Loss Non-Responder (*n* = 20) *	Weight Loss Responder (*n* = 22) *	*p*-Value **
**Weight Change (kg) from Baseline**	−2.0 (2.0)	−7.2 (2.5)	4.2 × 10^−9^
**% Weight Change from Baseline**	−2.1 (2.2)	−7.3 (2.2)	4.1 × 10^−9^
**Baseline Subject Characteristics**			
**Age (years)**	73.9 (3.9)	72.5 (3.9)	0.3
**Married (Yes)**	13 (65.0%)	14 (64.6%)	0.9
**Female**	16 (80.0%)	14 (63.6%)	0.2
**BMI (kg/m^2^)**	36.7 (4.5)	36.3 (6.06)	0.8
**Education: High School, No College**	4 (20.0%)	3 (13.6%)	0.7
**Income Less than $25,000/year**	4 (20.0%)	3 (13.6%)	0.7
**Waist-Hip Ratio**	0.93 (0.1)	0.93 (0.1)	0.9
**Gait Speed (s)**	1.05 (0.2)	1.06 (0.2)	0.8
**Grip Strength (kg)**	24.3 (7.62)	25.7 (11.9)	0.6
**30-Second Sit-to-Stand (repetitions)**	12.7 (3.63)	14.5 (7.19)	0.3
**Six-Minute Walk (m)**	372.0 (81.3)	403.6 (108.9)	0.3
**Depression (Yes)**	6 (30.0%)	6 (27.3%)	0.8
**Diabetes (Yes)**	6 (30.0%)	6 (27.3%)	0.8
**Fibromyalgia (Yes)**	6 (30.0%)	7 (31.8%)	0.9
**Hypertension (Yes)**	5 (25.0%)	12 (54.6%)	0.05
**Non-Skin Cancer (Yes)**	15 (75.0%)	15 (68.2%)	0.6
**Rheumatologic (Yes)**	7 (35.0%)	11 (50.0%)	0.3
**Stroke (Yes)**	9 (45.0%)	8 (36.4%)	0.5

* Mean (Standard Deviation) or Counts (Percent). ** *t*-test with Satterthwaite correction for unequal variances for continuous variables, chi-square for categorical variables, and Fisher’s Exact Test for categorical variables with small sample sizes (current smoker, education, income).

**Table 2 metabolites-13-00853-t002:** Mummichog pathway enrichment of responders vs. non-responders.

Pathway Number	Pathway Name	*p*-Value
1	Caffeine metabolism	0.000284
2	Valine, leucine, and isoleucine degradation	0.010064
3	Lysine metabolism	0.013518
4	Galactose metabolism	0.018886
5	Starch and Sucrose Metabolism	0.022688
6	Hexose phosphorylation	0.028288
7	Pentose phosphate pathway	0.039733
8	Arginine and Proline Metabolism	0.056501
9	TCA cycle	0.067128
10	Phytanic acid peroxisomal oxidation	0.067128
11	Beta-Alanine metabolism	0.085587
12	Fructose and mannose metabolism	0.094926
13	Glycosphingolipid metabolism	0.1043
14	Leukotriene metabolism	0.12449
15	Keratan sulfate degradation	0.12534

Full pathway enrichment is present in Appendix A

**Table 3 metabolites-13-00853-t003:** Caffeine and caffeine-related metabolites between responders and non-responders (all metabolites matched by RT, exact mass, and MS/MS).

Caffeine and Its Metabolites	Responder (Mean)	Non-Responder (Mean)	VIP *	*p*-Value **	Fold Change ***
1,3,7-trimethylxanthine	336,175.61	175,286.29	2.0	0.033	1.9
Theophylline	16,459.94	9,402.76	2.0	0.024	1.8
Paraxanthine	484,599.90	265,153.95	2.0	0.028	1.8
1-Methylxanthine	16,403.31	7,439.80	1.9	0.023	2.2
5-Acetylamino-6-amino-3-methyluracil	29,851.36	13,325.84	2.2	0.025	2.2
1,3-dimethyl uric acid	1,749.40	7,75.29	2.1	0.023	2.3
1,7-dimethyl uric acid	9,306.68	4,184.77	2.0	0.035	2.2

* VIP values were calculated by the OPLS-DA model in Figure 1. ** *p*-values were calculated using a two-sided *t*-test with the Satterthwaite correction for unequal variances. *** Positive fold changes denote compounds higher in responders, as compared to non-responders.

**Table 4 metabolites-13-00853-t004:** Additional metabolites differentiating responders and non-responders (*p* < 0.1) matched to in-house library of standards.

Compound	Ontology Level	Responder (Mean)	Non-Responder (Mean)	VIP	*p*-Value	Fold Change	Classification
N-Acetyl-Beta-Alanine	OL1	30,096	22,968	1.5	0.0448	1.3	Amino acids/peptides
L-Ornithine	OL1	51,267	44,420	1.6	0.0525	1.2	Amino acids/peptides
Dimethylglycine	OL1	93,750	69,088	2.4	0.0073	1.4	Amino acids/peptides
N6-Acetyl-L-Lysine	OL1	20,220	22,871	1.1	0.0873	−1.1	Amino acids/peptides
Methylcysteine	OL2a	130	184	1.4	0.0890	−1.4	Amino acids/peptides
N-Methyl-a-Aminoisobutyric Acid	OL2a	1,324,135	1,191,910	1.8	0.0640	1.1	Amino acids/peptides
Glycyl-Glutamate	OL1	49,370	36,909	1.3	0.0868	1.3	Amino acids/peptides
Glycyl-Serine	OL2a	170	296	1.5	0.0587	−1.7	Amino acids/peptides
Mevalolactone	OL2a	3,843	3,407	1.7	0.0788	1.1	Carbohydrate
Fucose	OL2a	3,118	2,623	2.0	0.0483	1.2	Carbohydrate
Xylose	OL2a	2,633	3,436	1.2	0.0718	−1.3	Carbohydrate
Galactitol	OL1	745	6,671	1.2	0.0531	−9.0	Carbohydrate
DL-Glyceraldehyde	OL1	428,309	516,829	1.7	0.0448	−1.2	Carbohydrate
Acetaminophen	OL1	311	766	1.5	0.0903	−2.5	Drug
Monoethyl Phthalate	OL2a	1,592	678	1.7	0.0273	2.3	Environmentally relevant compound
8-Hydroxyoctanoate	OL2a	77,724	68,981	1.8	0.0584	1.1	Lipids/Fatty acids
Glycerol	OL2a	79,047	68,602	1.8	0.0677	1.2	Lipids/Fatty acids
Octadecanoylcarnitine	OL1	19,143	23,878	1.7	0.0263	−1.2	Lipids/Fatty acids
Glycoursodeoxycholic Acid	OL1	63,971	25,373	1.5	0.0448	2.5	Lipids/Fatty acids
Dodec-2-Enedioic Acid	OL2a	7,871	6,438	1.6	0.0703	1.2	Lipids/Fatty acids
Palmitoylethanolamide	OL1	11,015	8,359	1.9	0.0164	1.3	Lipids/Fatty acids
Docosahexaenoate	OL1	19,544	14,054	1.7	0.0617	1.4	Lipids/Fatty acids
Adenosine	OL1	323	437	1.4	0.0235	−1.4	Nucleic acids
Cytosine	OL1	1,123	1,542	1.0	0.0967	−1.4	Nucleic acids
Pipecolate	OL1	126,578	42,799	1.6	0.0246	3.0	Phytochemical/microbiome-related
Pipecolinic Acid	OL1	717,629	582,803	2.0	0.0451	1.2	Phytochemical/microbiome-related
Dihydroferulic Acid	OL1	1,206	182	1.5	0.0671	6.6	Phytochemical/microbiome-related
3-(3-Hydroxyphenyl)-3-Hydroxypropanoic Acid	OL1	3,256	1,675	2.0	0.0253	1.9	Phytochemical/microbiome-related
3,4-Dimethoxyphenylpropanoic Acid	OL1	5,496	1,421	2.4	0.0031	3.9	Phytochemical/microbiome-related
3,5-Dihydroxybenzaldehyde	OL2a	1,697	812	1.8	0.0588	2.1	Phytochemical/microbiome-related
5-Hydroxypipecolic Acid	OL2a	664	956	1.6	0.0711	−1.4	Phytochemical/microbiome-related
3-Hydroxyhippuric Acid	OL1	8,526	3,771	1.8	0.0745	2.3	Phytochemical/microbiome-related
24,25-Dihydroxyvitamin D	OL2a	1,491	1,970	1.3	0.0833	−1.3	Vitamin

Ontology level (OL1) described in methods section. VIP values were calculated by the OPLS-DA model in Figure 1. *p*-values were calculated using a two-sided *t*-test with the Satterthwaite correction for unequal variances. Positive fold changes denote compounds higher in responders as compared to non-responders.

## Data Availability

Data is available upon request due to privacy.

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
