# Peer review of "Baseline Serum Biomarkers Predict Response to a Weight Loss Intervention in Older Adults with Obesity: A Pilot Study"

_metabolites, 2023, doi:10.3390/metabo13070853_

Round 1

Reviewer 1 Report

Dear Authors, thank you for an excellent manuscript. In the experiment section is mentioned that a PCA analysis was performed. How did the PCA scores separate the responders/non-responders samples - figure should be included in the manuscript. Other unsupervised methods tested - classification models? Caution using PLS-DA should be taken not to overinterpret model. Excellent that authors follow up with t-test with correction.

Reviewer 2 Report

This is an interesting and well-designed metabolomic study which needs some revisions before considering for acceptance

1. The Abstract is too long and it should be reduced including the most important findings of the study.

2. a) The Introduction is very poor and it should be enhriched by previous studies in this topic.

b) The authors should emphasize the risk factors of obesity in  older adults.

c) The authors introduce the precision medicine in general without specific recommendation in obesity and especially in the elderly.

d) Weight lost and sarcopenia is more frequently in the elderly than obesity. Please, explain and justify why obesity is also frequent in the elderly and what may cause in older adults.

e) More references should be added in the introduction.

3) Taking into consideration the sample size, the methodology and the results, the title of the article should include that this is a pilot study.

4) The Conclusion section should be enriched by the main findings of the study.

5) References list is very poor.

Minor editing of English language required

Reviewer 3 Report

Metabolites_2441936 Baseline serum biomarkers predict response to a weight loss intervention in older adults with obesity

This manuscript describes a metabolomics analysis between a small group of individuals that underwent a lifestyle intervention and where about 50% of the participants had more than 5% weight loss and the other 50% did not. The metabolomics analysis was performed on samples taken at baseline. I have a number of concerns with this study, that can be seen below.

1.     Methods, why were the data not corrected for multiple testing?

2.     Methods, why was dietary intake and/or physical activity at baseline not reported on and not evaluated as part of this analysis.

3.     Results, baseline metabolomics analysis: what were the results of the PCA analysis? It would be of interest to see this unsupervised analysis because it will indicate to the reader how much the groups differed from each other.

4.     Results, table 1: some of this data seems hyperspecific, I think that reporting age and BMI should be done to 1 decimal place, not two. It is unlikely that your data is that specific e.g. the scale and the measuring device for height are unlikely to be that specific. There should be units with gait speed/grip strength/sit to stands (in what time unit?)/six minute walk. There should also be information on the diagnostic cut-offs for depression/diabetes (type of diabetes)/fibromyalgia/hypertension/rheumatoid disease. Lastly, how many of the people with diabetes also had hypertension and/or stroke? Lastly, what medications were these participants taking and could this affect the metabolites present in the serum? Please also add the weight lost (both kg and %) to the table.

5.     Results, pathway analyses: Please explain why the pathway enrichment analysis was performed on all 9647 peaks in the untargeted dataset rather than the 697 significantly different peaks only (e.g. an investigation of the metabolic pathways enriched in the differentially different set rather than in all metabolites)? Did these analyses include both increased and reduced metabolites? Did you investigate the overall effect of the pathway at all?

6.     Results, pathway analyses: given that your analyses were not corrected for multiple testing, it is possible that most of the pathways are not representing true differences. Please add at least Q-values for the analyses to give the reader a better idea of the significance of the differences.

7.     Results, figure 3: what do the boxes and whiskers represent? What do the diamonds represent in these figures? There appears to be quite a spread in the levels of caffeine metabolites in each of the groups, is there anything that can explain this variability in the data? Are statistical analyses based on the normal distribution appropriate given the spread in the data, wouldn’t analyses based on non-normal distribution be better e.g. Kruskal-Wallis tests etc? It should be explained what VIP stands for and what it means in this context e.g. how does having a high VIP score make the metabolite more or less important.

8.     Results, Do the levels of caffeine metabolites correlate with the intake of caffeine in the participants? It would be of interest to investigate this to provide some clarity as to whether the differences in caffeine metabolites are due to different intake or different metabolism.

9.     Results, is the reduced amount of acetaminophen an indication that the responders actually were able to exercise more? Is there a correlation between the abundance of acetaminophen and the amount of exercise performed?

10.  Discussion: given the prominence of caffeine metabolism in the discussion, it is really important for the authors to link the caffeine metabolites to caffeine intake in this cohort. Given that the participants underwent both a dietary and a physical activity intervention, this data surely must be available.

Round 2

Reviewer 2 Report

The authors addressed all my suggestions and it has been considerably improved, meeting the criteria for publication.

Minor editing of English language required.